# A Novel Photocatalytic Functional Coating Applied to the Degradation of Xylene in Coating Solvents under Solar Irradiation

**DOI:** 10.3390/nano13030570

**Published:** 2023-01-31

**Authors:** Luying Sun, Yujie Tan, Hui Xu, Ruchen Shu, Zhi Liu, Ruina Zhang, Jianyuan Hou, Renxi Zhang

**Affiliations:** 1Institute of Environmental Science, Fudan University, Shanghai 200433, China; 2Foshan Shunde District Midea Washing Appliance Manufacturing Co., Ltd., Foshan 528311, China; 3Shanghai Institute for Design & Research on Environmental Engineering, Shanghai 200232, China

**Keywords:** dielectric barrier discharge, functional coating, VOCs, solar light photocatalyst

## Abstract

A novel photocatalytic functional coating was prepared with g-C_3_N_4_/TiO_2_ composites as the photocatalytic active component modified by dielectric barrier discharge (DBD), and it showed an efficient catalytic performance under solar light irradiation. The degradation of xylene released from fluorocarbon coating solvents by the g-C_3_N_4_/TiO_2_ composite coatings was investigated under simulated solar irradiation. The degradation efficiency of the coating mixed with DBD-modified 10%-g-C_3_N_4_/TiO_2_ showed a stable, long-lasting, and significantly higher activity compared to the coatings mixed with the unmodified catalyst. Ninety-eight percent of the xylene released from fluorocarbon coating solvents was successfully removed under solar light irradiation in 2 h. The properties of the catalyst samples before and after modification were evaluated using scanning electron microscopy (SEM), transmission electron microscopy (TEM), Fourier transform infrared spectroscopy (FT-IR), ultraviolet–visible (UV–vis) spectroscopy, X-ray photoelectron spectroscopy (XPS), and other characterization techniques. The results suggested that DBD-modified g-C_3_N_4_/TiO_2_ showed an improved capture ability and utilization efficiency of solar light with reduced band gap and lower complexation rate of electron–hole pairs. The prepared photocatalytic coating offers an environmentally friendly approach to purify the volatile organic compounds (VOCs) released from solvent-based coatings.

## 1. Introduction

Volatile organic compounds (VOCs) such as aromatic hydrocarbons, ketones, and esters released from commonly used solvent-based coatings have long been a major environmental concern [1,2]. In recent years, coatings with self-cleaning and air-purifying properties have received a great deal of attention. These functional coatings provide more diverse yet targeted functions for different application occasions without changing the original properties of the coating [3,4,5]. In particular, photocatalytic coatings utilize photocatalytic materials to degrade targeted pollutants under the irradiation of sunlight or visible light [6,7]. Among the available photocatalytic materials, TiO_2_ is the most widely applied material in photocatalytic functional coatings because of its excellent economic suitability, stable chemical properties, and high light utilization efficiency [8,9]. However, there are some critical challenges such as low quantum yield and weak solar light utilization efficiency [10]. To improve the efficiency of the catalyst, TiO_2_ is commonly integrated with other nanomaterials to increase its visible light response [11]. Noble metal elements are one of the most common doping species because they produce composites with higher photocatalytic activity and stability with a wider visible response range and stronger light absorption intensities [12,13]. However, doping with noble metals is expensive and can cause secondary pollution. The composition of TiO_2_ and nonmetallic elements has become an emerging trend as a solution to these problems [14,15,16].

As a narrowband semiconductor with a multilayered structure similar to graphene, g-C_3_N_4_ is currently an emerging material in the field of photocatalysis [17]. Its response to visible light, a band gap of approximately 2.7 eV, high chemical stability, and unique electronic properties make g-C_3_N_4_ an excellent nonmetallic photocatalyst [18,19]. The composition of TiO_2_ with g-C_3_N_4_ has received considerable attention for expanding the photocatalytic performance of semiconductor materials [20,21,22]. Currently, calcination [23,24], chemical vapor deposition [25], and atomic layer deposition [26,27] are the methods most commonly used to combine g-C_3_N_4_ with TiO_2_. However, these traditional methods tend to consume high energy and large amounts of chemicals and require long processing times. High-temperature calcination may even lead to the sintering of the active components in the catalyst, which reduces the activity of the composite to some extent [28]. In recent years, the dielectric barrier discharge (DBD) technology has proven to be a promising method for catalyst synthesis and modification. It enables reactions to be easily carried out under mild conditions without the addition of chemicals, thereby offering the advantages of low energy consumption and low environmental impact [29]. Many studies have reported that TiO_2_-based composite photocatalysts modified by DBD can exhibit excellent surface properties and enhanced absorption of visible and infrared light [30,31,32]. Further, the reaction preparation process can be significantly simplified and the catalyst can be quickly prepared.

This study reports a novel photocatalytic functional coating with DBD-modified g-C_3_N_4_/TiO_2_ as the active component and investigates its properties and performance in the degradation of xylene released from the coating solvent under simulated solar light conditions. The mechanism of xylene degradation by the photocatalytic coating is also described.

## 2. Experimental Methods

### 2.1. Materials

Nanometer-scale titanium dioxide (99.8%, 150 nm, anatase) was purchased from Shandong Tian Xing New Materials Co., Ltd., Shandong, China. Nanometer-scale graphite-phase carbon nitride (99%, 1–10 μm) was purchased from Jiangsu Xian Feng Nanomaterials Technology Co., Ltd., Jiangsu, China. Both materials were analytical grade and used as received. A fluorocarbon varnish was purchased from Jiangsu Can Wei Chemical Co. Ltd., Jiangsu, China.

### 2.2. Preparation of Photocatalysts

First, TiO_2_ and g-C_3_N_4_ nanoparticles with various mass ratios (0%, 3%, 5%, 7%, and 10% g-C_3_N_4_) were mixed to yield samples of 1 g each and dispersed in 40 mL deionized water. The samples were ultrasonic for 30 min to ensure homogeneous mixing, completely dried in an oven at 85 °C, cooled to room temperature, and ground into powder.

The mixed samples and quartz cotton were placed in a DBD reactor with a coaxial cylindrical structure (Figure 1) and treated for 1 h. The DBD-modified composites were denoted as g-C_3_N_4_/TiO_2_, whereas the unmodified mixture of g-C_3_N_4_ and TiO_2_ was denoted as un-g-C_3_N_4_/TiO_2_. The modification treatment system included a gas supplement, an AC power supply system, and a DBD discharge reaction chamber; ammonia (14.7 mol% of air) was supplied as the atmospheric gas. A mass flow controller (Horiba Stec-4400, Kyoto, Japan) was used to control the inlet flow of NH_3_ to 60 mL/min. The DBD discharge reaction chamber included an inner tube (quartz glass, outer diameter of 3 mm, wall thickness of 0.8 mm), outer tube (quartz glass, outer diameter of 12 mm, inner diameter of 10 mm, wall thickness of 1 mm), inner electrode (inner tube filled with stainless steel powder), and outer electrode (a copper foil sheet wrapped around the outer wall of quartz glass, thickness of 0.3 mm, length of 15 mm). The frequency of the DBD discharge chamber was measured using a 200 MHz digital fluorescent oscilloscope (Tektronix, TDS2024B, Beaverton, OR, USA), and the applied voltage and power were measured using a voltage Lissajous plot. The external electrode was discharged at a sinusoidal voltage of 5 kV, a frequency of 20 kHz, and a discharge power of 34 W.

### 2.3. Characterization

An X-ray powder diffractometer (Ultima IV, Rigaku, Japan) was used to record X-ray diffraction (XRD) patterns to investigate the crystal structure of the composites. Scanning electron microscopy (SEM, S-4800, Hitachi, Japan) and transmission electron microscopy (TEM, JEM-2100F, JEOL, Tokyo, Japan) were used to investigate the morphology of the samples. A Fourier transform infrared spectroscope (IRTracer-100, Shimadzu, Japan) with KBr as a reference was used to record the Fourier transform infrared (FT-IR) spectra of the samples. A surface area and porosity analyzer (ASAP 2460, Quantachrome, Tallahassee, Florida, USA) was used to analyze Brunauer–Emmett–Teller (BET). X-ray photoelectron spectroscopy (XPS, K-Alpha, Thermo Scientific, Waltham, MA, USA) was used to analyze the elemental composition and valence variations of the samples. An ultraviolet–visible (UV–vis) spectrophotometer (UV-3600, Shimadzu, Kyoto, Japan) was used to perform UV–vis diffuse reflectance spectroscopy (DRS). A UV/V/near-infrared (NIR) fluorescence spectrometer (FLS980, Edinburgh, UK) was used to record the photoluminescence (PL) spectra of the samples at an excitation wavelength of 325 nm. A three-electrode cell was used to measure the photocurrent and perform electrochemical impedance spectroscopy (EIS) with Ag/AgCl and Pt lines as the reference and counter electrodes, respectively. An electron paramagnetic resonance spectrometer (EMXplus, Bruker, Karlsruhe, Germany) was used to perform electron paramagnetic resonance (EPR) analysis, with N,N-dimethylpyrroline N-oxide (DMPO) as the trapping agent for ·OH and ·O_2_^−^ at room temperature. Specifically, a 1 mg/mL aqueous solution sample was prepared and sonicated for 5 min. Subsequently, 30 μL of this sample was mixed with 30 μL of DMPO for ·OH detection. The same procedure was followed to detect ·O_2_^−^ using a methanol solution instead of the aqueous solution.

### 2.4. Preparation of Coatings

Quartz sheets measuring 20 mm × 20 mm × 1 mm were mechanically polished, cleaned, and dried. Then, 0.05 g samples of a mixture of TiO_2_ and g-C_3_N_4_, pure TiO_2_, and different ratios of modified g-C_3_N_4_/TiO_2_ catalysts were each added to 1 g of the fluorocarbon varnish. The samples were then mixed and sonicated for 30 min, followed by mechanical stirring to ensure uniform dispersion of the catalysts in the coatings. Then, 0.02 g samples of the mixed coatings were uniformly coated on the surface of the quartz sheet and dried in an incubator at a certain temperature. A series of photocatalytic coating films were prepared in a similar manner.

### 2.5. Photoactivity Evaluation

The photocatalytic performance of the coatings was evaluated at room conditions (20 ± 1 °C and 50% relative humidity) under simulated solar light. The duration of light for each test set was 2 h. A 250 mL quartz chamber was used as the photocatalytic reactor, and a 350 W xenon lamp was used as the light source to simulate solar light. The reactor was filled with air, and the reaction was carried out at atmospheric pressure. Xylene was used as the targeted degradation substance. Before starting the experiment, the photocatalytic coating films were placed in the reactor. A certain volume of saturated vapor on the surface of the xylene solution was injected into the reactor under dark conditions. The reactor was left in the dark for 0.5 h. Irradiation began when the concentration of xylene in the reactor reached an adsorption equilibrium of approximately 220 ppm. The concentration of xylene in the reactor was recorded to analyze the degradation efficiency using a gas chromatograph (GC-9896) equipped with a Heyasep column, flame ionization detector (FID), and thermal conductivity detector (TCD) after every 15 min of irradiation during the experimental process. The conversion of the xylene was calculated as follows: (1)$${\mathrm{Xylene}}_{\mathrm{conversion}}=\frac{C_{t0}-C_{t}}{C_{t}}\times 100\%$$
where C_t0_ refers to the initial concentration of xylene, and C_t_ is the concentration of xylene at time t.

## 3. Results and Discussion

### 3.1. XRD Analysis

The XRD patterns were used to evaluate the crystallographic composition and structure of the modified catalyst samples, as shown in Figure 2. For the various ratios of DBD-modified g-C_3_N_4_/TiO_2_ composites, the pronounced diffraction peaks were consistent with the pure TiO_2_ sample, indicating that the synergistic effects of g-C_3_N_4_ and ammonia plasma treatment did not significantly affect the anatase structure (JCP-DS21-1272) of TiO_2_ [33,34]. The existence of the (100) and (002) crystal planes of the graphite material (JCPDS, Card No. 87-1526) g-C_3_N_4_ was confirmed by the presence of the pronounced diffraction peaks at 2θ values of 13.0 and 27.5°. No characteristic peak of g-C_3_N_4_ can be found in the XRD patterns of the series of composite samples, most likely due to the low content of g-C_3_N_4_ on the surface of TiO_2_. 

### 3.2. Molecular and Physical Characterization

SEM was used to analyze the surface morphology of the pure TiO_2_, un-g-C_3_N_4_/TiO_2_, and g-C_3_N_4_/TiO_2_ composites. TiO_2_ was uniformly distributed as irregularly spherical nanoparticles, as shown in Figure 3a. The relatively distinguishable nanosheet-like structure of g-C_3_N_4_ deposited on the TiO_2_ surface can be observed, as shown in Figure 3b,c. Compared to un-g-C_3_N_4_/TiO_2_, the DBD-modified g-C_3_N_4_/TiO_2_ composite samples tended to aggregate into small clusters, causing the surface to be rough with numerous mesoporous structures, which possibly resulted from the electron etching effect from the DBD treatment [35].

The microscopic morphologies of pure TiO_2_, g-C_3_N_4_, un-g-C_3_N_4_/TiO_2_, and g-C_3_N_4_/TiO_2_ samples were analyzed using TEM, as shown in Figure 4. The (101) characteristic lattice of anatase TiO_2_ is shown in Figure 4b, and the layered structure of g-C_3_N_4_ is presented in Figure 4c. As shown in Figure 4d,e, TiO_2_ and g-C_3_N_4_ were homogeneously mixed, and no obvious connection interface can be observed. As shown in Figure 4f,g, a portion of the g-C_3_N_4_/TiO_2_ composite surface shows a uniform deposition plane compared with Figure 4a, indicating the uniform integration of g-C_3_N_4_ with the TiO_2_ surface. As shown in Figure 4g, the surface of g-C_3_N_4_ changed from smooth to rough and exhibited the characteristic (002) lattice of graphite. The TiO_2_ surface exhibited an amorphous conversion after DBD treatment (Figure 4g). A closely packed interface with a fuzzy appearance forms a hybridized structure between TiO_2_ and g-C_3_N_4_, implying a surface connection between the two substances. This hybridized structure has extensive interfacial contacts and provides more active sites for the catalyst, thereby contributing to improved photocatalytic activity [19].

### 3.3. FT-IR Analysis

The functional groups on the surfaces of the samples were characterized using FT-IR spectroscopy, as shown in Figure 5. TiO_2_ exhibited a broad absorption peak at the wave center of 3388 cm^−1^, which can correspond to the O-H stretching pattern of water adsorbed on the TiO_2_ surface, and a narrower absorption peak at 1628 cm^−1^, which corresponds to the O-H bending pattern of surface water. A region of strong absorption peaks can be observed below 830 cm^−1^, which corresponds to Ti-O-Ti bond absorption [36]. The active hydroxyl groups on the surface of TiO_2_ inhibit photogenerated electron–hole complexation and simultaneously react with photogenerated holes to produce hydroxyl radicals (·OH), thereby improving the photocatalytic performance of the modified samples [37]. For g-C_3_N_4_, there was a broad absorption peak corresponding to the stretching pattern of the N-H bond at the wave center of 3067 cm^−1^, and a range of typical absorption peaks in the range of 1237–1640 cm^−1^, which correspond to the stretching pattern of the C=N double bond and C-N bond on the carbon–nitrogen ring. The absorption peak at 806 cm^−1^ corresponds to the breathing mode of the triazine units [38]. For the series of composites, it is evident that their spectra are consistent with those of TiO_2_ and un-g-C_3_N_4_/TiO_2_, indicating that the functional groups of the major component TiO_2_ in the sample did not change. However, the typical absorption peak of g-C_3_N_4_ was not detected in the spectra of the composites, which may be attributed to the low content of g-C_3_N_4_ in the samples and the fact that the DBD modification treatment caused little change in the functional groups of the samples.

### 3.4. Nitrogen Sorption

The specific surface areas and pore characteristics of the samples were investigated by analyzing the N_2_ adsorption–desorption isotherms and corresponding pore size distribution curves of the samples, shown in Figure 6, as well as the corresponding surface details obtained from the adsorption isotherms, shown in Table 1. Generally, a material with a larger specific surface area provides more activation sites for the rapid migration of products; thus, increasing the specific surface area becomes one of the most important methods for enhancing the catalytic activity of the material [39]. As shown in Figure 6a, all the un-10%-g-C_3_N_4_/TiO_2_ and modified composites demonstrate a type IV adsorption isotherm with typical hysteresis lines, indicating the presence of a mesoporous/microporous structure in the samples [40]. Additionally, the DBD treatment did not alter the natural structure of the material. As shown in Figure 6b, most of the pore size range of pure TiO_2_, un-10%-g-C_3_N_4_/TiO_2_, and modified composite samples was concentrated below 10 nm, implying the presence of mesopores and micropores in the catalysts. As shown in Table 1, the BET specific surface areas of g-C_3_N_4_, TiO_2_, un-10%-g-C_3_N_4_/TiO_2_, 3%-g-C_3_N_4_/TiO_2_, 5%-g-C_3_N_4_/TiO_2_, 7%-g-C_3_N_4_/TiO_2_, and 10%-g-C_3_N_4_/TiO_2_ are 5.24, 305.80, 255.11, 117.18, 212.34, 129.57, and 129.24 m^2^/g, with corresponding pore volumes of 0.018, 0.355, 0.202, 0.239, 0.347, 0.318, and 0.237 cm^3^/g, respectively. Because the specific surface area of g-C_3_N_4_ was insignificant, a large amount of g-C_3_N_4_ was embedded in the TiO_2_ interstices during the integration process, thereby reducing the specific surface area of un-10%-g-C_3_N_4_/TiO_2_ and the composites. For different proportions of the modified composites, 5%-g-C_3_N_4_/TiO_2_ had a greater BET specific surface area than the other samples. As 5%-g-C_3_N_4_/TiO_2_ had the largest total pore volume V_pore_, more mesopores were newly generated, contributing to the increase in its BET specific surface area. Specifically, the total pore volume V_pore_ increased from 0.202 cm^3^/g for the un-10%-g-C_3_N_4_/TiO_2_ to 0.237 cm^3^/g for the 10%-g-C_3_N_4_/TiO_2_ composites, which can facilitate the diffusion of xylene molecules within the material to some extent. Based on these results, the change in the specific surface area was not a major factor contributing to the improvement in the photocatalytic performance of this composite.

### 3.5. XPS Analysis

XPS was used to analyze the surface elements of the samples. The complete spectra of TiO_2_, un-10%-g-C_3_N_4_/TiO_2_, and DBD-modified 10%-g-C_3_N_4_/TiO_2_ are shown in Figure 7a. The distinct binding energy peaks of C, O, and Ti were detected near 284.56 eV (C1s), 529.98 eV (O1s), and 458.26 eV (Ti2p), respectively. From the spectra of un-10%-g-C_3_N_4_/TiO_2_ and 10%-g-C_3_N_4_/TiO_2_, the binding energy peak of N was further identified near 399.07 eV(N1s). Specifically, Figure 7b shows three peaks in the C1s area of pure TiO_2_ at binding energies of 284.80, 286.08, and 288.30 eV, which may correspond to sp^3^ hybridized carbon [41] C-O and C=O bonds [42]. Particularly, the C=O bond may be the result of the adsorption of the photolysis product CO and the adsorption of CO_2_ in the air. Three peaks of DBD-modified 10%-g-C_3_N_4_/TiO_2_ at binding energies of 284.38, 285.88, and 287.88 eV may correspond to the C-C, C-N, and C=N bonds [43,44], respectively, demonstrating the existence of g-C_3_N_4_ in the sample. Considering the atomic ratio of C1s, the C content of g-C_3_N_4_/TiO_2_ decreased from 28.71% to 25.87% after DBD modification, which confirmed that carbon atoms can be activated to an excited state under ammonia DBD treatment. Further, it was demonstrated that DBD treatment caused electron etching on the TiO_2_ surface.

The fitted O1s spectra shown in Figure 7c exhibit three peaks, with the peak at 529.48 eV corresponding to the lattice oxygen of TiO_2_, including O-Ti^4+^, and the peaks at 530.98 and 532.38 eV corresponding to oxygen vacancies or reactive oxygen groups, such as O-H and surface-adsorbed species [45]. It has been reported that surface oxygen-containing species play a vital role in the degradation of contaminants [46]. The area ratio of the surface oxygen-containing species increased from 16.36% to 17.58% after DBD modification, indicating that DBD introduced more surface oxygen-containing species to promote the degradation process.

As shown in Figure 7d, the two strong peaks located at 458.20 and 463.94 eV in the spectra of Ti2p correspond to Ti2p^3/2^ and Ti2p^1/2^ [47], respectively. These peaks of Ti2p of 10%-g-C_3_N_4_/TiO_2_ shift toward a lower binding energy compared to pure TiO_2_ and un-10%-g-C_3_N_4_/TiO_2_. This phenomenon demonstrates an increase in Ti^3+^ [48], indicating the introduction of Ti^3+^ by DBD treatment. As shown in Figure 7e, the N1s spectra show three peaks at binding energies of 398.25, 400.06, and 401.18 eV, which may correspond to C-N=C [49], N-(C)_3_, and C-N bonds [50,51], respectively, further demonstrating the existence of g-C_3_N_4_.

### 3.6. Characterization of Optical Properties

The UV–vis DRS spectra of pure TiO_2_, un-10%-g-C_3_N_4_/TiO_2_, and a series of modified composite samples are shown in Figure 8a. All samples show a strong absorption of short-wavelength light below 420 nm, whereas g-C_3_N_4_/TiO_2_ composites were more or less red-shifted compared to pure TiO_2_ and un-10%-g-C_3_N_4_/TiO_2_, producing the absorption of visible light as well as an enhanced absorption of UV light. In combination with the spectra, the band gaps of the samples were obtained using the truncation method [52], as shown in Figure 8b. If the energy of the photon equals or exceeds the band gap of the semiconductor, the electrons in the valence band absorb the photon and enter the conduction band, producing electron–hole pairs [53]. As shown in Figure 8b, the pure TiO_2_ and un-10%-g-C_3_N_4_/TiO_2_ samples have a band gap of 3.30 eV, and the DBD-modified g-C_3_N_4_/TiO_2_ samples all show reduced band gaps, of which 10%-g-C_3_N_4_/TiO_2_ has the narrowest of 3.17 eV. The narrower band gap of the modified composite allows for a higher transfer efficiency of photogenerated carriers, facilitating the absorption of photons from the valence band into the conduction band, which results in more electron–hole pairs [54]. Consequently, the utilization of solar light by the catalyst was enhanced, thereby strengthening its photocatalytic activity.

The PL spectra were used to determine the migration and recombination processes of the photogenerated electron pairs, and the results are shown in Figure 9. During the test, the TiO_2_ sample emitted too weakly to maintain the same slit and was unable to produce an accurate signal in the spectra. The PL fluorescence intensities at approximately 440 and 460 nm are as follows in descending order: un-10%-g-C_3_N_4_/TiO_2_ > 7%-g-C_3_N_4_/TiO_2_ > 3%-g-C_3_N_4_/TiO_2_ > 10%-g-C_3_N_4_/TiO_2_ > 5%-g-C_3_N_4_/TiO_2_. The un-10%-g-C_3_N_4_/TiO_2_ exhibited the highest peak because the material has a high complexation rate of photogenerated electron–hole pairs. The PL signal intensity of the 7%-g-C_3_N_4_/TiO_2_ composite showed a little decrease compared to un-10%-g-C_3_N_4_/TiO_2_, which was consistent with the result of UV–vis DRS spectra. The significant decrease in the PL peaks of the other proportion of DBD-modified composites implies a reduction in the complexation rates of the photogenerated electrons and holes, indicating a high photocatalytic activity due to a well-established contact between g-C_3_N_4_ and TiO_2_. In addition, the large reduction in the PL peak of 10%-g-C_3_N_4_/TiO_2_ exhibited a higher photogenerated electron transfer efficiency compared to un-10%-g-C_3_N_4_/TiO_2_, indicating that the DBD treatment can significantly enhance the photocatalytic activity of the composites.

### 3.7. EIS Analysis

The charge-transfer efficiency at the sample interface was investigated using EIS, and the results are shown in Figure 10. Compared with TiO_2_ and un-g-C_3_N_4_/TiO_2_, the radius of the arc in the EIS Nyquist curve plots of the DBD-modified catalyst electrodes was significantly smaller, representing a significant decrease in the resistance at the modified catalyst interface, which greatly improved the charge-transfer efficiency [55]. In particular, 10%-g-C_3_N_4_/TiO_2_ has the smallest arc radius and the highest charge-transfer efficiency. Therefore, the interfacial interaction between g-C_3_N_4_ and TiO_2_ after the DBD treatment effectively accelerated the charge transfer, thereby facilitating the effective separation of the photogenerated electrons and holes.

### 3.8. Photoactivity Test

To investigate the photoactivity of the photocatalytic coatings in the degradation of xylene, irradiation for 2 h under simulated solar light was performed, and the degradation efficiency was calculated using Equation (1); the results are shown in Figure 11a. For the unloaded fluorocarbon resin coating, insignificant degradation of xylene occurred within 2 h, indicating negligible photocatalytic activity of the single fluorocarbon resin coating under solar irradiation. In contrast, the photocatalyst-loaded coating exhibited a promising photocatalytic performance under the same conditions. The TiO_2_-loaded coating film achieved a degradation efficiency of 57% for xylene after 2 h of irradiation, while the un-10%-g-C_3_N_4_/TiO_2_-loaded coating degraded approximately 45% of xylene. The reaction rate of the un-10%-g-C_3_N_4_/TiO_2_-loaded coating was faster than that of TiO_2_ in the first 45 min because the composite mixed with g-C_3_N_4_ gained a larger absorption range of light and enhanced utilization of sunlight. However, the activity of the un-10%-g-C_3_N_4_/TiO_2_-loaded coating was significantly weaker than that of TiO_2_ after 45 min, probably because of the higher amount of g-C_3_N_4_ covering the active sites of TiO_2_ as well as the reduced specific surface area. Notably, the plasma-modified g-C_3_N_4_/TiO_2_ coating films showed a far superior photocatalytic activity compared to the TiO_2_ coating film, and the 10%-g-C_3_N_4_/TiO_2_ coating film showed the most significant performance, removing 98% of xylene under solar light irradiation. Meanwhile, the degradation rate of the photocatalytic coating was apparently higher in the first 30 min than in the subsequent stages owing to the reduced concentration of xylene in the later stages of the reaction. 

The DBD treatment enhanced the efficiency of the catalyst under solar light irradiation by reducing the band gap. Concurrently, the effective compounding of TiO_2_ with g-C_3_N_4_ also promoted the efficient separation of the photogenerated electron–hole pairs. A moderate integration of g-C_3_N_4_ and DBD treatment had synergistic effects on improving the absorption and conversion efficiency of xylene by photocatalytic coatings. The 10%-g-C_3_N_4_/TiO_2_ coating film was tested over five cycles (Figure 11b), and the results show no significant decrease in its photocatalytic activity for the degradation of xylene, proving the coating film has a long service life. To further assess the applicability of the prepared coating, a performance comparison test was carried out between the 10%-g-C_3_N_4_/TiO_2_ coating and a commercial high-functional coating (purchased from SKSHU Paint Co., Ltd., FuJian, China) under the same conditions and methods, as shown in Figure 11c. The results show that the reaction processes for the degradation of xylene under solar light irradiation were similar between both coatings. The 10%-g-C_3_N_4_/TiO_2_ coating achieved a degradation rate of 96% for xylene, slightly higher than the 91% rate achieved by the commercial coating, and both showed significant photocatalytic activity.

As mentioned above, the DBD-modified g-C_3_N_4_/TiO_2_-loaded photocatalytic coating has a high photocatalytic degradation capability under solar light irradiation, which not only broadens the light absorption wavelength of the TiO_2_ coating film but also improves the photocatalytic efficiency over the pure TiO_2_ coating film. In addition, the dominant methods of preparing photocatalysts inevitably involve the addition of solvents and high-temperature calcination, which pose problems such as secondary pollution and high energy consumption, and thus are considered environmentally unfriendly options. This study employed a DBD method to prepare photocatalysts that achieved acceptable degradation efficiency with a green approach that consumed less energy and was economically feasible.

### 3.9. DMPO Spin-Trapping EPR Spectra

EPR spectroscopy was used to investigate the effects of plasma modification and g-C_3_N_4_ on the formation of active oxides of TiO_2_. The presence of ·OH and ·O_2_^−^ in the system was detected in deionized water and methanol, respectively, using DMPO as the free-radical trapping agent; the results are shown in Figure 12. The signals of ·OH and ·O_2_^−^ were detected in both the pure TiO_2_ and 10%-g-C_3_N_4_/TiO_2_ samples, with the EPR peak of the 10%-g-C_3_N_4_/TiO_2_ sample being slightly stronger than that of pure TiO_2_. It can be concluded that both composites produced high levels of active oxygen species under simulated solar light irradiation for xylene removal; however, the efficient transfer of photogenerated carriers via the heterojunction formed during the plasma treatment played a more significant role.

### 3.10. Mechanism

A possible degradation mechanism of xylene on the DBD-modified g-C_3_N_4_/TiO_2_ photocatalytic coating under simulated solar light is described. First, the fluorocarbon varnish traps and adsorbs gaseous pollutants, accelerating the removal efficiency of xylene to some extent. In addition, DBD treatment produces more mesopores and disordered structures on the catalyst surface, facilitating the removal of gaseous pollutants. Second, the absorption of light by the catalyst increases owing to the reduced band gap. It is evident from the XPS spectra that the DBD treatment promotes the formation of Ti^3+^ three-dimensional orbitals in TiO_2_, and the presence of Ti^3+^ creates surface defects in the active sites, which improve the absorption of light below the conduction band, facilitating the migration and separation of electrons and holes generated by light [48]. Meanwhile, the charge transfer between TiO_2_ and g-C_3_N_4_ under solar light irradiation is shown in Figure 13. g-C_3_N_4_ in the modified catalyst is excited to produce electron–hole pairs by capturing photons under solar light irradiation. It has been reported that the conduction band (CB) side of g-C_3_N_4_ is higher than that of TiO_2_, while the valence band (VB) side of TiO_2_ is higher than that of g-C_3_N_4_, and a preferable electron medium (RGO) exists between g-C_3_N_4_ and TiO_2_, resulting in a preferred flow of photogenerated electrons from the CB of g-C_3_N_4_ into the CB of TiO_2_ instead of into the VB of TiO_2_. These processes facilitate the separation of photogenerated electron–hole pairs [56]. The adsorbed oxygen on the surface of g-C_3_N_4_ can easily trap the electrons in its CB side to form ·O_2_^−^ radicals, whereas OH^−^/H_2_O can react with the holes in the VB of TiO_2_ to form ·OH radicals, both of which turn to a charge balance and are reactive oxygen species that play a significant role in the photocatalytic degradation of xylene. The above discussion demonstrates that the surface properties of the DBD-treated catalysts facilitate the removal of free gaseous pollutants, and the redox capacity of the modified catalysts integrated with g-C_3_N_4_ is also improved, both of which synergistically enhance the photocatalytic activity.

## 4. Discussion

In this study, a novel photocatalytic functional coating using low-temperature DBD-modified g-C_3_N_4_/TiO_2_ as the active component was successfully prepared. The photocatalytic functional coating exhibited an exceptional performance in the degradation of xylene under simulated solar light conditions. The analytical results show that the photocatalytic activity of the active components in the coating under solar light induction was significantly enhanced by the synergistic effect of the DBD treatment and g-C_3_N_4_ integration. Ammonia DBD treatment generated more surface oxygen vacancies and reduced the band gap of TiO_2_, leading to improved optical properties of the catalyst effective separation and transfer of photogenerated carriers, and thus enhanced the efficiency of the material in capturing and utilizing solar light. In conclusion, the successful preparation of the photocatalytic coating provides an efficient, low-carbon, and environmentally friendly approach for purifying VOCs in paint solvents.

## Figures and Tables

**Figure 1 nanomaterials-13-00570-f001:**
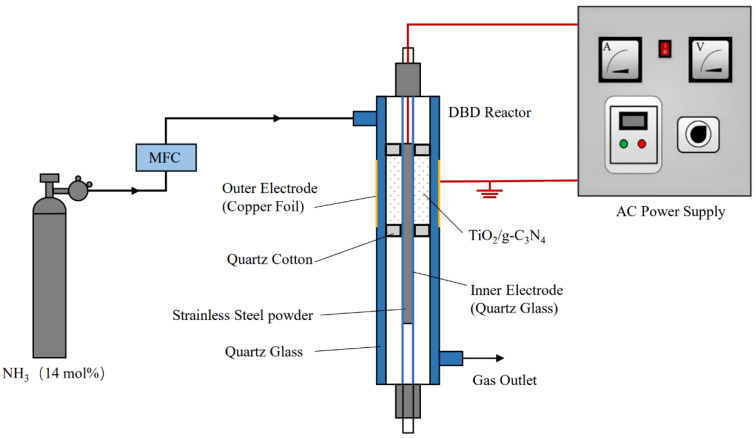
Schematic of the ammonia DBD-modified system.

**Figure 2 nanomaterials-13-00570-f002:**
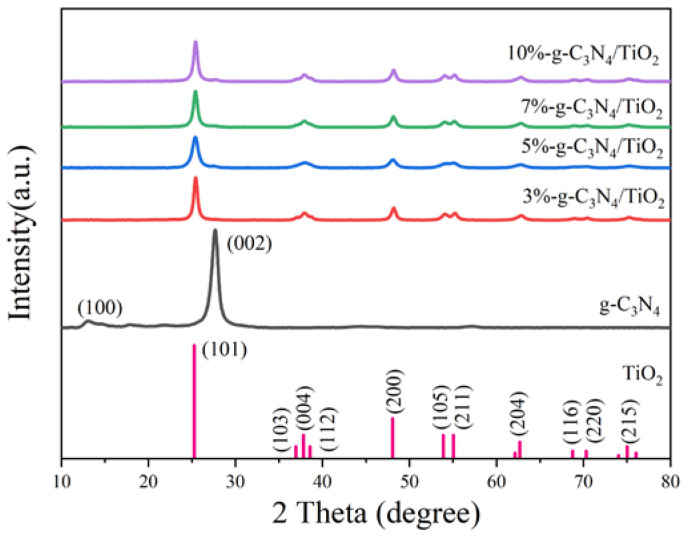
X-ray diffraction patterns of the samples.

**Figure 3 nanomaterials-13-00570-f003:**
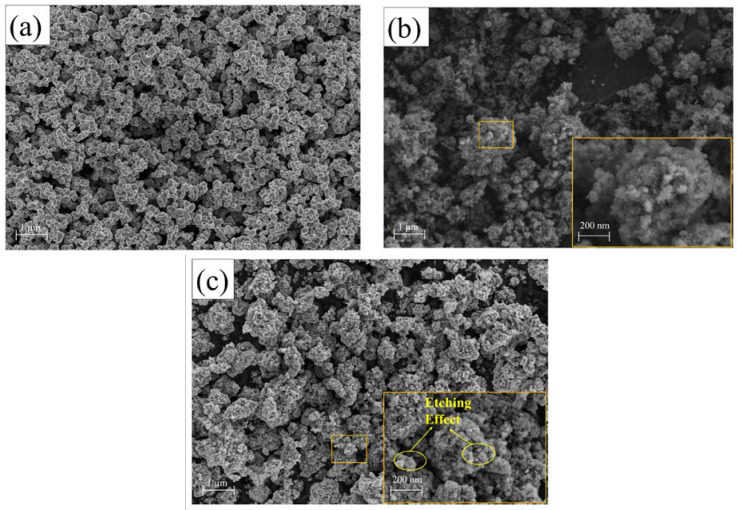
Scanning electron microscopy images of (**a**) pure TiO_2_, (**b**) un-g-C_3_N_4_/TiO_2_, and (**c**) g-C_3_N_4_/TiO_2_ composite.

**Figure 4 nanomaterials-13-00570-f004:**
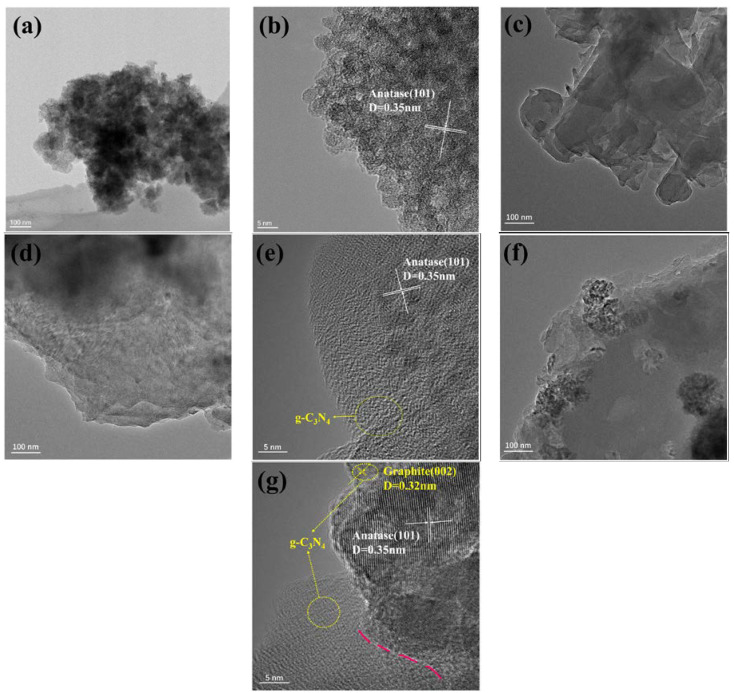
Transmission electron microscopy images of (**a**,**b**) TiO_2_, (**c**) g-C_3_N_4_, (**d**,**e**) un-g-C_3_N_4_/TiO_2_, and (**f**,**g**) g-C_3_N_4_/TiO_2_ composite.

**Figure 5 nanomaterials-13-00570-f005:**
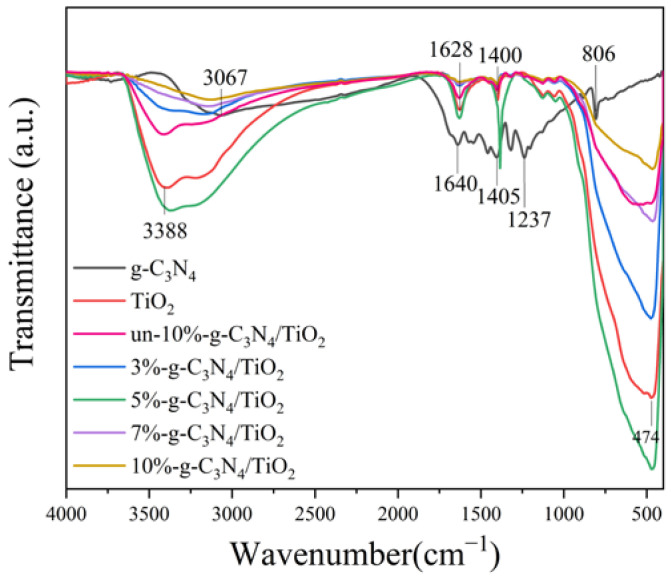
Fourier transform infrared spectra of the samples.

**Figure 6 nanomaterials-13-00570-f006:**
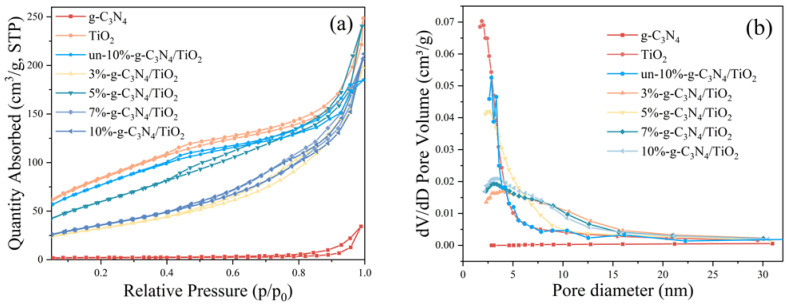
(**a**) Nitrogen adsorption–desorption isotherms and the (**b**) corresponding pore size distribution curves of TiO_2_, g-C_3_N_4_, un-10%-g-C_3_N_4_/TiO_2_, and the series of composites.

**Figure 7 nanomaterials-13-00570-f007:**
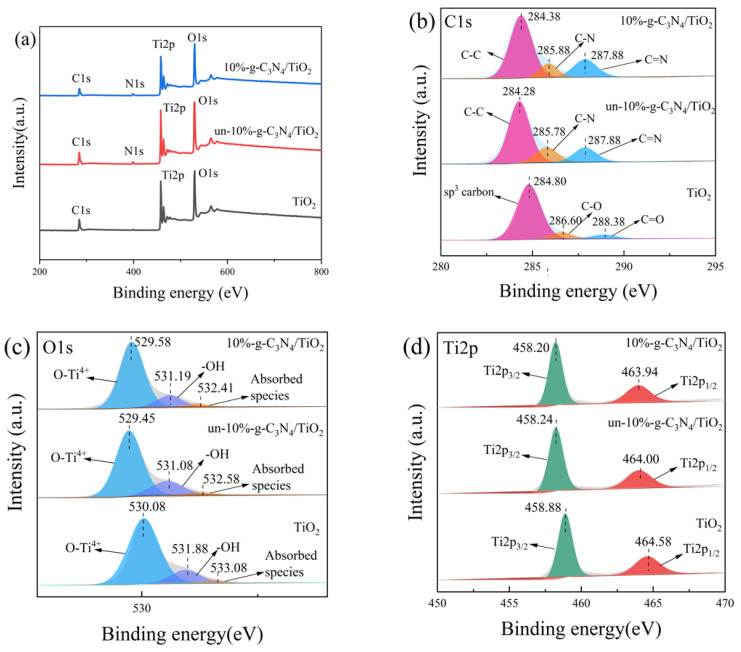
X-ray photoelectron spectra of TiO_2_, un-10%-g-C_3_N_4_/TiO_2_, and 10%-g-C_3_N_4_/TiO_2_ samples: (**a**) survey, (**b**) C1s, (**c**) O1s, (**d**) Ti2p, and (**e**) N1s.

**Figure 8 nanomaterials-13-00570-f008:**
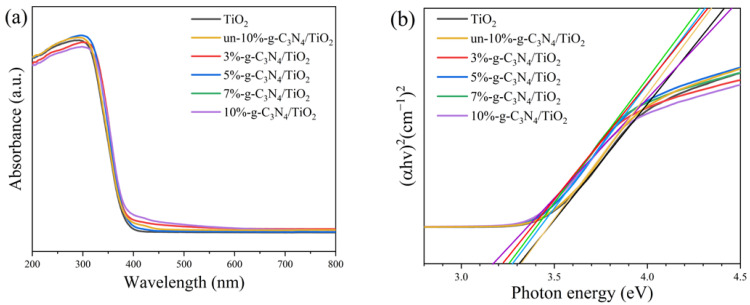
(**a**) UV–vis diffuse reflectance spectra and the (**b**) corresponding band gaps of the samples.

**Figure 9 nanomaterials-13-00570-f009:**
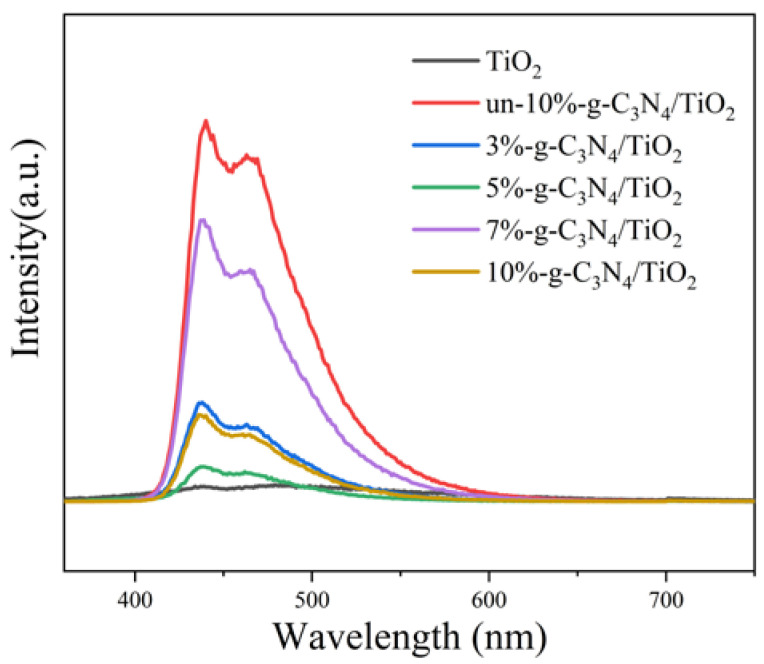
Photoluminescence spectra of the samples.

**Figure 10 nanomaterials-13-00570-f010:**
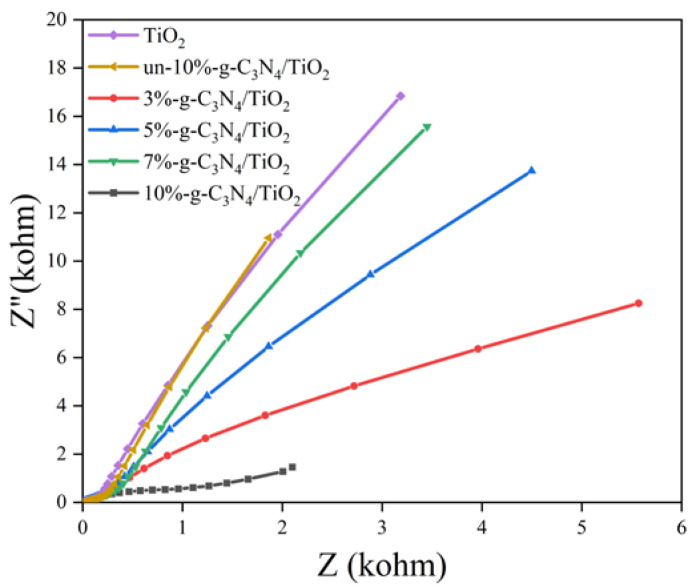
EIS Nyquist plots of the samples.

**Figure 11 nanomaterials-13-00570-f011:**
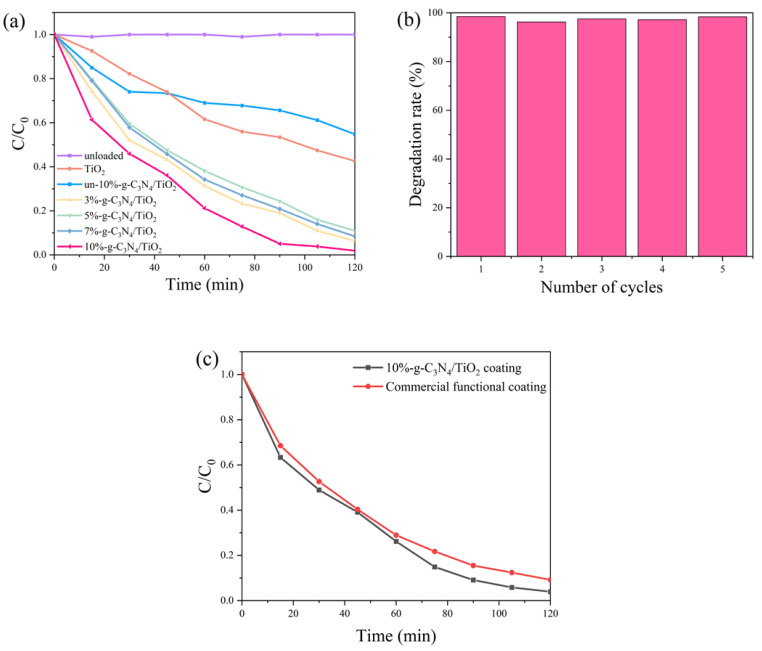
(**a**) Xylene degradation of the photocatalytic coatings under simulated solar irradiation, (**b**) the recycle numbers of the 10%-TiO_2_/g-C_3_N_4_-loaded coating film, and (**c**) a comparison between the prepared coating and a commercial functional coating.

**Figure 12 nanomaterials-13-00570-f012:**
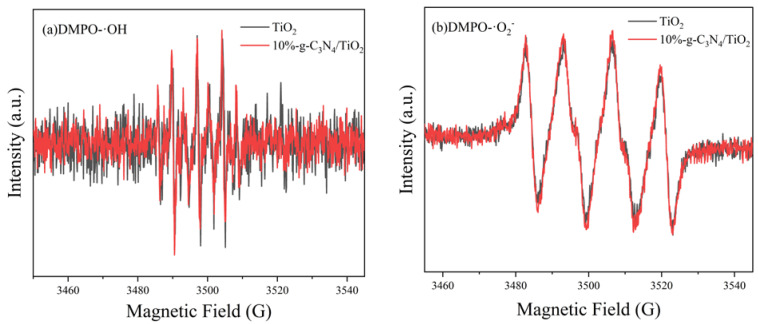
Electron paramagnetic resonance spectra of TiO_2_ and 10%- g-C_3_N_4_/TiO_2_ after 5 min solar light irradiation: (**a**) DMPO-·OH; (**b**) DMPO- ·O_2_^−^.

**Figure 13 nanomaterials-13-00570-f013:**
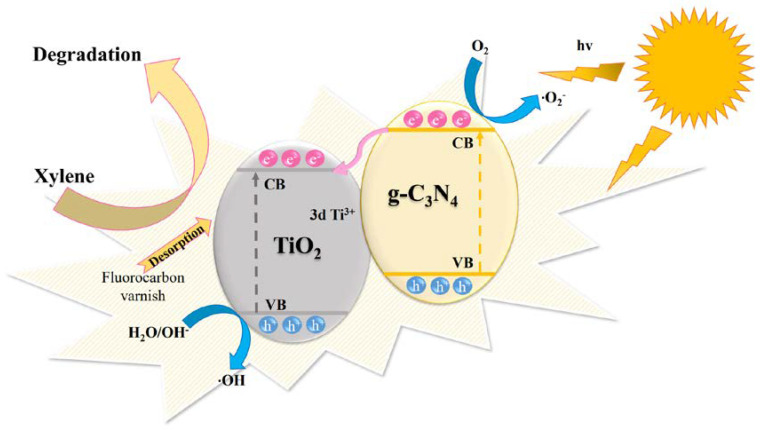
Xylene degradation mechanism for the charge transfer between TiO_2_ and g-C_3_N_4_ under solar light irradiation.

**Table 1 nanomaterials-13-00570-t001:** Physical properties of TiO_2_, g-C_3_N_4_, and the modified photocatalyst samples.

Sample	S_BET_ (m^2^/g)	V_pore_ (cm^3^/g)	Pore Size (nm)
g-C_3_N_4_	5.24	0.018	21.5919
TiO_2_	305.80	0.355	5.6994
un-10%-g-C_3_N_4_/TiO_2_	255.11	0.202	4.4808
3%-TiO_2_/g-C_3_N_4_	117.18	0.239	8.0502
5%-TiO_2_/g-C_3_N_4_	212.34	0.347	7.8124
7%-TiO_2_/g-C_3_N_4_	129.57	0.318	9.7446
10%-TiO_2_/g-C_3_N_4_	129.24	0.237	7.0663

## Data Availability

The data presented in this study are available on request from the corresponding author. The data are not publicly available due to privacy.

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
