# Peer review of "A Novel Photocatalytic Functional Coating Applied to the Degradation of Xylene in Coating Solvents under Solar Irradiation"

_nanomaterials, 2023, doi:10.3390/nano13030570_

Round 1

Reviewer 1 Report

Sun et al. describe the preparation, characterization and testing of TiO2/g-C3N4 photocatalysts incorporated into a fluorocarbon varnish.  Methods used to characterize the material included FT-IR, XPS, X-ray powder diffraction, SEM, UV-vis spectroscopy, electrochemical impedance spectroscopy, BET adsorption and XPS.  The TiO2/g-C3N4 nanoparticles of composition 0-10% were treated in dielectric barrier discharge(DBD) reactor.  Results indicated significant increase in the rate of decomposition of xylene under simulated solar radiation.

Use of carbon nitride(g- C3N4) as a photocatalyst has been known for over a decade.  The enhanced performance of the treated catalysts is an interesting finding, though its utility in fluorocarbon coatings is questionable due to their being phased out.  On the whole the manuscript in its present form is well-written with many experimental details including a thorough description of the DBD reactor. However, the Discussion section is entirely too short and more effort is needed explaining trends in their results.  For example, the authors should comment on results in Table 1, Figure 9 and Figure 11a.  Why does the 5% sample has such a greater surface area than 3% and 7%?  What explanation is there for the absence of any apparent trend in the photoluminescence data? Are these results reproducible? Some information on experimental methods is still needed as noted below.

Figure 13 needs a more extensive caption summarizing the mechanism shown.  There are technical issues to address with the figure.  Unpaired electrons on O2- and OH are not clearly seen.  There is a missing hydrogen atom (H+?) and a charge imbalance.   Ordinarily, the number of holes and conductance band electrons would be the same. 

Neither FTIR or XRD indicated the presence of C3N4.   XPS does provide evidence supporting its existence, but a TiO2 control spectra is needed.

Minor points:

Figure 9: Axis label—Intensity

In the photocatalytic reactor, indicate the gases present(air or oxygen) and its concentration or pressure.  Were there multiple injections of xylene?

Author Response

Thank you so much for your high efficiency in reviewing our manuscript entitled “A novel photocatalytic functional coating applied to the degradation of xylene in coating solvents under solar irradiation”. Those comments are all valuable and very helpful for revising and improving our article. We have studied carefully and the main corrections in the paper and the responds to your comments are as follows:

Point 1: The discussion section is entirely too short and more effort is needed explaining trends in their results. For example, the authors should comment on results in Table 1, Figure 9 and Figure 11a.  Why does the 5% sample have such a greater surface area than 3% and 7%?  What explanation is there for the absence of any apparent trend in the photoluminescence data? Are these results reproducible?

Response 1: Thanks for your professional comments. We have expanded the discussion section as you suggested. The additional comment on results in Table 1 was added in section 3.4. The additional comment on results in Figure 9 was added in section 3.6. Some little errors were also revised in L-318. For the results of Figure 11a, each set of the experiments has been repeated two or three times and the results were stable, and the results of the cyclic experiment shown in Figure 11b also maintained at one level, so these results are reproducible.

Point 2: Figure 13 needs a more extensive caption summarizing the mechanism shown. There are technical issues to address with the figure. Unpaired electrons on O2- and OH are not clearly seen.  There is a missing hydrogen atom (H+?) and a charge imbalance. Ordinarily, the number of holes and conductance band electrons would be the same.  

Response 2: Thanks for your suggestion. The caption of Figure 13 has been revised to appropriately summarize the mechanism shown. And Figure 13 has been replaced with a new version to address the technical issues you professionally pointed out. Unpaired electrons on O2- and OH now can be seen clearly and electron-hole pairs have been aligned. As shown in text of section 3.10, the adsorbed oxygen was reduced by the electrons in CB of g-C3N4 to ·O2radicals, while OH/H2O was oxidized by the holes in VB of TiO2 to ·OH radicals, which achieved to a charge balance, and there is no H+ in this system. The relevant contents have been added to the text of section 3.10.

Point 3: Neither FTIR or XRD indicated the presence of C3N4. XPS does provide evidence supporting its existence, but a TiO2 control spectra is needed.  

Response 3: Thanks for your professional thinking. The TiO2 control spectra was provided in Figure 7 to stress the presence of g-C3N4 in un-10%-g-C3N4/TiO2 and 10%-g-C3N4/TiO2 samples, and the relevant comment on the results was added to the text of section 3.5.

Point 4: Minor points: Figure 9: Axis label—Intensity

Response 4: Thanks for your comment. The spelling error of axis label has been corrected.

Point 5: In the photocatalytic reactor, indicate the gases present (air or oxygen) and its concentration or pressure. Were there multiple injections of xylene?

Response 5: Thanks for your suggestion. The experimental information was added in section 2.5. Only one injection of xylene was given before the start of the experiment as its initial concentration, and there were not multiple injections during the test processes.

Reviewer 2 Report

In my opinion, the way of preparing the catalytic system and the way of its application is interesting and should be published. The authors also comprehensively characterized the morphology of the catalytic system they prepared. However, the most important feature of the catalyst is its activity. The activity assessment presented by the authors does not allow to assess the actual activity of the catalyst in comparison to commercially available photocatalysts. This problem can be solved with a simple comparison experiment. I suggest you perform such an experiment and attach the results.
Other perceived problems.
1. Is the varnish used hydrophilic? If not, how did electron holes react with water?
2. The text and Figure 13 show that the use of the system significantly reduces the energy needed to excite the catalyst. Experimental data (Fig. 8) do not confirm the existence of such a difference between the tested catalysts.
3. L-76 anatase?
4. L-138 xylene is it a product or a substrate?
5. L-146 Were the samples irradiated for only 15 minutes? It's unclear.

Author Response

Thank you so much for your high efficiency in reviewing our manuscript entitled “A novel photocatalytic functional coating applied to the degradation of xylene in coating solvents under solar irradiation”. Those comments are all valuable and very helpful for revising and improving our article. We have studied carefully and the main corrections in the paper and the responds to your comments are as follows:

Point 1: The activity assessment presented by the authors does not allow to assess the actual activity of the catalyst in comparison to commercially available photocatalysts. This problem can be solved with a simple comparison experiment. I suggest you perform such an experiment and attach the results.

Response 1: Thanks for your professional suggestion. We performed a comparison experiment between the prepared photocatalytic coating and one commercial photocatalytic coating purchased from SKSHU. The performance test results were attached in Figure 11c, and some comments on the results were added in section 3.8.

Point 2: Is the varnish used hydrophilic? If not, how did electron holes react with water?  

Response 2: Thanks for your thinking. There are hydrophilic groups in the varnish we use, so water can be attracted to the surface of the coating and react with electron holes.

Point 3: The text and Figure 13 show that the use of the system significantly reduces the energy needed to excite the catalyst. Experimental data (Fig. 8) do not confirm the existence of such a difference between the tested catalysts.  

Response 3: Thanks for your professional comments. Our intention was to demonstrate that the composite of TiO2 and g-C3N4 can promote the charge transfer in the prepared catalyst by those characterization results, and we mainly elaborated this process in the mechanism section (3.10). Some revisions were made to Figure 13 to explain the process clearly.

Point 4: L-76 anatase?

Response 4: Thanks for your comment. It was one written error and was deleted.

Point 5: L-138 xylene is it a product or a substrate?

Response 5: Thanks for your thinking. Xylene is the target degradable and the relative content was corrected.

Point 6: L-146 Were the samples irradiated for only 15 minutes? It's unclear.

Response 6: Thanks for your reminding. The samples were totally irradiated for 2h and the concentration of xylene was recorded at every 15 min-irradiation during the reaction process. Some experimental information was added in section 2.5 to make the methods clear.

Round 2

Reviewer 2 Report

I am satisfied with the improvements made by the authors in the manuscript. Therefore, in my opinion, the manuscript can be published.